# HER2-Positive Gastroesophageal Cancers Are Associated with a Higher Risk of Brain Metastasis

**DOI:** 10.3390/cancers14235754

**Published:** 2022-11-23

**Authors:** Gary Tincknell, Asma Naveed, Jane Nankervis, Ayesha Mukhtiar, Ann-Katrin Piper, Therese M. Becker, Lorraine Chantrill, Morteza Aghmesheh, Kara Lea Vine, Marie Ranson, Daniel Brungs

**Affiliations:** 1Illawarra Health and Medical Research Institute, University of Wollongong, Wollongong, NSW 2522, Australia; 2Illawarra Cancer Care Centre, Illawarra Shoalhaven Local Health District, Wollongong, NSW 2500, Australia; 3School of Chemistry and Molecular Biosciences, University of Wollongong, Wollongong, NSW 2522, Australia; 4School of Medicine, University of Wollongong, Wollongong, NSW 2522, Australia; 5NSW Health Pathology, Wollongong, NSW 2522, Australia; 6Southern IML, Wollongong, NSW 2500, Australia; 7Ingham Institute for Applied Medical Research, Liverpool, NSW 2170, Australia; 8UNSW Medicine, University of New South Wales, Kensington, NSW 2052, Australia; 9School of Medicine, Western Sydney University, Sydney, NSW 2560, Australia

**Keywords:** oesophageal adenocarcinoma, brain metastasis, human epidermal growth factor 2, survival, gastroesophageal cancer, gastric adenocarcinoma

## Abstract

**Simple Summary:**

For patients who develop brain metastasis, the consequences can be devastating, with neurological complications and potentially rapid death. Early intervention with surgery and radiotherapy can prevent neurological complications and improve patient survival. Gastroesophageal cancers which express human epidermal growth factor 2 (HER2) behave differently to those lacking expression. We aimed to review patients who developed brain metastasis from gastroesophageal cancers and assess the risk of HER2 expression on developing brain metastasis. A total of 12 patients developed brain metastasis from the 201 assessed with metastatic gastroesophageal cancer. The life expectancy of patients with gastroesophageal cancer was similar regardless of HER2 expression. However, the risk of developing brain metastasis was significantly increased. Clinicians need to have a low threshold for investigation of neurological symptoms in patients with HER2-positive gastroesophageal adenocarcinoma. The assessment of current HER2-targeted treatments, and development of new drugs is needed in this under researched area of gastroesophageal cancers.

**Abstract:**

Brain metastasis from gastroesophageal adenocarcinomas (GOCs) is a rare but a devastating diagnosis. Human epidermal growth factor receptor 2 (HER2) is a prognostic and predictive biomarker in GOCs. The association of HER2 with GOC brain metastasis is not known. We performed a retrospective analysis of patients with GOCs with known HER2 status between January 2015 and November 2021. HER2 was assessed on either the primary tumour or metastasis by immunohistochemistry or in situ hybridization. The diagnosis of brain metastasis was made on standard imaging techniques in patients with symptoms or signs. HER2 results were available for 201 patients, with 34 patients (16.9%) HER2 positive. A total of 12 patients developed symptomatic brain metastasis from GOCs, of which 7 (58.3%) were HER2 positive. The development of symptomatic brain metastasis was significantly higher in the HER2-positive GOCs (OR8.26, 95%CI 2.09–35.60; *p* = 0.0009). There was no significant association of HER2 status and overall survival in patients with brain metastasis. Although the rate of brain metastasis remains low in GOCs, the incidence of symptomatic brain metastasis was significantly higher in patients with HER2-positive tumours.

## 1. Introduction

Gastric, gastroesophageal junction and oesophageal adenocarcinomas (GOCs) are aggressive and lethal cancers. Combined, these tumours accounted for 8.7% of global new cancer cases and 13.2% of cancer-related deaths in 2020 [1]. Adenocarcinomas account for approximately 90% of gastric cancers [2,3], and 80% of oesophageal cancers in Western countries [4]. The incidence of oesophageal adenocarcinoma is increasing due to dietary changes and increased body habitus [5]. Adenocarcinomas arising from the stomach, gastroesophageal junction and oesophagus share molecular subtypes and act in a similar fashion, and therefore can be considered a single clinical entity [6]. Despite improvements in cancer treatments and the development of targeted therapies, GOCs continue to have a dismal 5-year overall survival [2,7].

The identification of metastasis is a devastating event, rendering the disease process incurable, and a switch to palliative intent treatment. GOCs commonly metastasise to distant lymph nodes, the liver, peritoneum, and lungs [8,9]. Secondary brain metastasis from GOCs is relatively rare, occurring in 1.9% and 6% of patients with stage IV gastric and oesophageal adenocarcinomas, respectively [9,10]. Risk factors for developing metastasis include tumour depth of invasion at the primary site, invasion of the lymphatic or vascular system, an infiltrative growth pattern and poorly differentiated tumours [11,12].

The development of brain metastases is inherently catastrophic. Local treatments, such as surgical resection and radiotherapy, provide survival benefit for patients [13,14]. However, the neurological complications associated with intracranial malignancy and associated treatments negatively impact patients’ quality of life such as fatigue, headaches, neurocognitive decline, and focal neurology [15,16]. Early detection and treatment of intracranial disease result in improved post-intervention status [17].

Chemotherapy remains the backbone of standard-of-care therapies for patients with advanced GOCs [18,19]. Fluoropyrimadines, platinums, taxanes and topoisomerase I inhibitors have all been shown to improve overall survival in metastatic GOCs [20]. Furthermore, recent phase III data have demonstrated the additional benefits of immunotherapy with checkpoint inhibitors such as Nivolumab and Pembrolizumab, both in combination with chemotherapy and as monotherapy [21,22,23,24]. Clinical biomarkers in routine practice are the human epidermal growth factor receptor 2 (HER2 or *ERBB-2* receptor), program death-ligand 1 (PD-L1) and DNA mismatch repair genes [22,25,26].

Trastuzumab is a monoclonal antibody targeting HER2, preventing signalling mediated by the HER2 through the phosphatidylinositol 3-kinase (PI3K) and mitogen-activated protein kinase (MAPK) cascades [27]. HER2 overexpression occurs in 6–30% of GOCs [28,29,30,31]. The addition of trastuzumab to a cytotoxic therapy for the treatment of metastatic GOCs improves overall survival by 2.7 months compared to standard of care [29]. Other HER2-directed therapies (e.g., lapatanib and trastuzumab emtansine) have been assessed in clinical trials of metastatic GOCs but none have yet demonstrated improved survival over chemotherapy with trastuzumab [32,33]. Ongoing trials of combination blockade of HER2 and immunotherapy have shown promising objective response rates of the tumours, but survival data are awaited [34]. Novel agents such as trastuzumab deruxtecan are showing promise in the treatment of GOCs with improved radiological response rates and survival after multiple lines of prior treatment [35,36]. None of these agents have assessed their effects on intracranial outcomes for GOC patients with brain metastasis.

Brain metastasis from GOCs is managed in a multimodal manner using surgery and radiotherapy. However, chemotherapy and trastuzumab have poor permeability across the blood–brain barrier and systemic treatments therefore have limited benefit to patients with brain metastases [37,38].

Given the limited information and rarity of this disease, the current study evaluates the incidence of symptomatic brain metastases in patients with GOC adenocarcinomas and explores the association with HER2 status.

## 2. Materials and Methods

We performed a retrospective review of patients treated for GOCs within the Illawarra Shoalhaven Local Health District (ISLHD), New South Wales, Australia. Patients were identified from the New South Wales cancer registry and from electronic medical records at ISLHD. Patients treated from January 2015 to November 2021 were included in the analysis.

Patients were included if they were treated for cancers originating in the oesophagus, gastroesophageal junction, or stomach. We only included patients diagnosed with adenocarcinomas (including subtypes) and excluded alternative pathologies such as squamous cell carcinoma (SCC) and gastrointestinal stromal tumour (GIST). Patients were included if they met the criteria for HER2-targeted therapies as outlined by the pivotal ToGA trial, that being patients with inoperable locally advanced, recurrent or metastatic adenocarcinoma [29]. Patients were excluded from brain metastasis assessment if their intracranial disease was thought to be related to cancers from alternative origin based on the opinion of the treating oncologist (e.g., one patient had historic gastric cancer, but active metastatic lung cancer).

Staging and identification of metastatic disease were as per routine care. Staging was performed with standard investigations (e.g., computer tomography and fluorodeoxyglucose-positron emission tomography) by the treating physician and documented in the electronic medical records; no retrospective assessments of imaging were performed to include or exclude metastasis in patients. Specifically, metastasis to the brain was diagnosed using standard imaging techniques including computer tomography (contrast) and magnetic resonance imaging and documented by the reporting radiologist. Given the rarity of brain metastases in GOCs, routine brain imaging is not performed. Patients were only investigated for brain metastasis in the presence of clinical symptoms and signs suggestive of intracranial pathology, such as headaches, seizures, nausea or focal neurological symptoms. Non-symptomatic patients were not screened for asymptomatic brain metastasis.

### 2.1. HER2 Detection

As part of standard therapy, HER2 immunohistochemistry (IHC) and in situ hybridization (ISH) were performed on patients who demonstrated locally advanced or metastatic disease in line with standard clinical practice. HER2 assessment occurred on either primary tumour or metastatic disease depending on the diagnostic tissue available. As per Australian guidelines and funding restrictions, only patients with metastatic disease were tested. We identified (*n* = 58) patients with metastatic disease who did not have a documented HER2 status as part of their standard-of-care treatment. In these patients, HER2 IHC was performed on archival formalin-fixed, paraffin-embedded tissue and scored by experienced anatomical pathologists (authors AN, JN and AM).

Local practice dictates that for patients with HER2 IHC scores 2+ or 3+ (HER2 staining of more than 10% of the cells), the HER2 status was confirmed using ISH analysis. Standard HER2 IHC scoring reporting is: 0—no HER2 staining; 1—less than 10% HER2 membranous staining of cancer cells; 2—more than 10% of cells staining weak/moderate cancer cells; 3—more than 10% of cells showing complete staining of tumour cells [39]. HER2 IHC 0 and 1+ were considered negative, 2+ equivocal and 3+ positive. For patients with HER2 IHC reported as equivocal, we arranged for confirmation with ISH using the Ventana HER2 Dual ISH DNA probe cocktail. Appendix A shows representative HER2 IHC and ISH results. All HER2 testing was performed using routine assays in National Association of Testing Authorities (NATA) certified pathology laboratories and reported by trained pathologists.

Only patients with definitive HER2 results were included in the final analysis. A total of 8 patients were excluded due to insufficient or unavailable tissue for HER2 testing.

### 2.2. End Points

Primary endpoint was to assess if HER2 expression in patients with symptomatic brain metastasis increased the likelihood of developing brain metastasis.

Secondary endpoints of this analysis were overall survival and survival following symptomatic brain metastasis. Overall survival was assessed from date of diagnosis to date of death. Survival following brain metastasis diagnosis was calculated from date of diagnosis to death or censoring.

### 2.3. Statistics Tools

Statistical analysis was performed using R (version 4.0.2). Descriptive summary statistics were compared using a chi squared test for categorical variables and one-way ANOVA for continuous variables. Odds ratios were tested using Fisher’s exact method. Survival analysis was performed using Kaplan–Meier and Cox proportional models [40]. *p* values less than 0.05 were considered significant for all assessments.

### 2.4. Ethics

This project was approved by South Western Sydney Local Health District (Australia) HREC (N°: HREC/15/LPOOL/121). A waiver of consent was granted to access health records.

## 3. Results

We identified 201 patients who met the criteria of unresectable and progressive, recurrent, or metastatic GOCs with a HER2 result. Of these, 125 patients presented with de novo metastatic disease; 54 patients had curative intent surgery and recurrence during follow up; 22 had locally advanced disease and had non-surgical management only. A total of 34 (16.9%) patients were HER2 positive. HER2-positive tumours were diagnosed by ISH in 21 cases and by IHC in the remaining 13. A total of 167 (83.1%) patients were HER2 negative. HER2-negative GOCs were confirmed by ISH in 47 cases, and by IHC in 120 cases.

Patient characteristics are described in Table 1.

Twelve (6.0%) patients had symptomatic brain metastases. Patients with symptomatic brain metastasis were more likely to be younger (*p* = 0.005) and have a primary tumour in the gastroesophageal junction. HER2-positive patients were more likely to have symptomatic brain metastases than HER2-negative patients (20.6% vs. 3.0%; *p* < 0.0004). The odds ratio (OR) for developing symptomatic brain metastasis in the HER2-positive versus HER2-negative population was highly significant at 8.26 (95%CI 2.09–35.60; *p* = 0.0009).

The median follow-up time was 46.5 weeks (range 0 to 331 weeks). There was no statistically significant difference in the median overall survival for patients with symptomatic brain metastasis, 50.0 weeks (IQR 20.1–94.3), and without brain metastasis, 46.2 weeks (IQR 14.5–89.6, *p* = 0.5; Figure 1).

A total of 8 (66.7%) of the 12 patients with symptomatic brain metastases presented with metastatic disease at diagnosis, 3 had recurrence following previous resection of the primary disease and the final patient had progression following non-surgical treatment of their locally advanced disease. Of the patients who had symptomatic brain metastasis, nearly half (41.6%) had brain metastasis at presentation. There was no significant difference in time from diagnosis to the diagnosis of symptomatic brain metastasis between HER2-positive and -negative GOCs (median = 19.3 weeks vs. 18.6 weeks *p* = 1.0).

Although there were only 12 patients with brain metastasis, we observed no statistical difference in survival following the diagnosis of symptomatic brain metastasis in HER2-positive or -negative disease (median 29.9 weeks vs. 22.9 weeks; *p* = 0.6).

The clinical course and treatments of the patients with brain metastases from GOCs are summarised in Table 2. There is a wide range in overall survival seen in this cohort (range 6–902 days). The patients with the longest survival had minimal extracranial disease with small-volume nodal disease only at time of diagnosis of brain metastases. Both patients (50- and 52-year-old males) presented with brain metastasis from primary gastroesophageal junction tumours, with only local nodal metastasis as their other sites of metastatic disease. Representative images of their initial brain metastases can be seen in Figure 2. This permitted adequate local treatment for the brain metastasis, including surgical metastectomy and post-operative radiotherapy, prior to proceeding to systemic therapies (see Table 2 below for further details). Interestingly, the first site of subsequent relapse in both patients was intracranial. This suggests a reduced intracranial effect of the systemic treatments used. Of the 12 patients, only 5 (41.7%) were able to proceed to systemic treatments following the diagnosis of brain metastasis. This demonstrates the devastating effect that brain metastasis has on functional status.

## 4. Discussion

Brain metastases arising from gastric, gastroesophageal junction and oesophageal adenocarcinomas (collectively GOCs) are rare but devastating diagnosis for patients. We have identified HER2 as a key risk factor for developing symptomatic brain metastasis in patients with GOCs. In our cohort, the likelihood of developing symptomatic brain metastasis was significantly higher than the HER2-negative group (OR 8.26). To the best of our knowledge, this is the first time the increased likelihood of developing brain metastasis in HER2-positive GOCs has been demonstrated for patients. HER2-associated brain metastasis has been previously described in patients with mixed underlying tumour histology (including squamous cell carcinoma), isolated tumour site (i.e., stomach), all gastrointestinal sites of primary tumours [41,42,43]. The issue with these studies is they did not assess all patients eligible for HER2-directed therapies. The studies were unable to advise on the increased risk of developing brain metastasis in HER2-positive GOCs. This finding has immediate clinical utility, highlighting the importance of HER2 assessment in all patients with metastatic GOCs. These patients should have a lower threshold for assessment of the presence of brain metastasis. Patients who have their brain metastasis identified at an earlier stage often have more treatment options, and this is in part due to less complex surgery and a better baseline function. Additionally, these patients experience less morbidity associated with their disease and treatment.

In our single-centre analysis, we showed a brain metastasis rate of 6.0% in metastatic GOCs. This is a slightly higher rate than Cavanna et al. [42], which showed a rate of 2.33% in the gastric cancer population, but similar to the 6.3% reported by Hejleh et al. in all histology oesophageal cancer [41]. In our cohort of patients, the site of the primary malignancy was mostly the oesophagus or gastroesophageal junction, probably accounting for a prevalence just lower than Hejleh et al.

As primary adenocarcinomas from the stomach, gastroesophageal junction and oesophagus are molecularly similar, then these tumours can safely be analysed together [6]. Although previous studies have shown brain metastasis can occur in HER2-positive GOCs, they have not been able to describe the risk of developing metastasis [41,42]. The additional benefit of this study is that we have demonstrated the increased risk of developing brain metastasis for HER2-positive cancers in the whole population of non-resectable locally advanced, recurrent and metastatic GOCs.

Brain metastasis in our cohort only occurred in patients with clinical stage III (nodal-positive disease) or metastatic disease at presentation. This is consistent with the findings of Hejlah, where seven of the nine resected patients who developed brain metastasis had nodal-positive disease at diagnosis (no patients in this study had metastatic disease at presentation). All bar one patient who developed brain metastasis were male in our study; brain metastasis are more common in male patients, and the rate of brain metastasis identification was similar to other GOCs cohorts [44].

Historically, the only management of cerebral metastasis has been surgery and radiotherapy, with chemotherapy having minimal intracranial effect. Although only 12 patients were identified with brain metastasis, our results show the importance of including adequate local treatments (intracranial surgery and radiotherapy) in the management of patients with brain metastasis.

In metastatic HER2-positive GOCs, the ToGA trial revolutionised treatment by the introduction of biomarker (HER2)-directed therapy (trastuzumab) in combination with chemotherapy. Despite the improvements in survival for HER2-positive GOCs overall, trastuzumab is a large molecule that fails to cross the blood–brain barrier [37]. The intracranial effect is minimal when administered via a systemic route. Trastuzumab can be administered directly into the cerebrospinal fluid via surgically placed reservoirs or intrathecal injections; however, no clinical trials have assessed the outcomes in GOCs [45,46]. In GOCs, other HER2-directed therapies have not derived clinical benefit to date.

In addition to GOCs, HER2 is an established biomarker for breast cancer. Evidence in patients with breast cancer demonstrates that HER2-positive breast cancers had higher rates of brain metastasis [47,48,49]. HER2 expression appears to be an important pathway in the development of brain metastasis from HER2-expressing tumours. In this case, adjuvant trastuzumab resulted in delayed time to brain metastasis formation [50]. The use of pertuzumab (recombinant monoclonal IgG1 antibody) in metastatic breast cancer also showed delayed time to brain metastasis formation [51]. Upon development of HER2-associated breast cancer with brain metastasis, single-agent trastuzumab improved patient survival despite its apparent inability to cross the blood–brain barrier [47]. Pertuzumab/Paclitaxel, and trastuzumab-emtansine regimens showed improved overall response rates, progression-free survival and overall survival [47]. The novel agent trastuzumab deruxtecan (humanised monoclonal antibody anti-human HER2-topoisomerase I inhibitor payload conjugate) shows promise, with significant overall response rates and progression-free survival benefits for patients with brain metastasis from HER2-positive breast cancer in a single-arm phase II trial [52]. Lapatinib is a small-molecule tyrosine kinase inhibitor (TKI) against epidermal growth factor receptor and HER2. Surprisingly, lapatinib did not demonstrate any intracranial benefits for HER2-positive breast cancer patients in spite of its increased ability to cross the blood–brain barrier [47]. Other small TKI agents, neratinib, tucatinib and pyrotinib, show intracranial effects on HER2-positive brain metastasis from breast cancers in early phase trials [47]. As discussed in the introduction, several of these agents have been assessed in GOCs without similar clinical efficacy, whilst several trials are currently ongoing.

The recent progression of novel biomarker-targeted therapies (e.g., immunotherapy) has transformed a number of cancer treatments including GOCs [21,22,23]. In melanoma, immunotherapy has potentially removed the necessity of surgery in the management of brain metastasis [53]. There have not been any trials demonstrating the effectiveness of immunotherapy on brain metastasis in GOCs.

Due to the low number of brain metastasis in GOCs, large trials have proven difficult to develop to clarify the optimum treatment paradigm for these patients.

### Limitations of the Study

This single-centre retrospective study is limited by the low number of brain metastasis. As such, with this in combination with 6 of the 12 patients dying within 3 months from diagnosis of brain metastasis, we were unable to tease out any survival benefit obtained from systemic HER2-directed therapies; this represents the aggressive nature of this disease. Nevertheless, we have demonstrated the rate of brain metastasis to be significantly higher in patients with HER2-positive GOCs than those HER2-negative GOCs.

## 5. Conclusions

We have demonstrated that HER2-positive GOCs are more likely to develop brain metastasis. Whilst survival of patients with brain metastasis is similar between HER2-positive and -negative populations, the clinical consequences of brain metastasis for GOC patients can be devastating. Prompt diagnosis of brain metastasis can result in earlier intervention with the goal of minimising morbidity. We advocate for the early screening of patients who have neurological symptoms and HER2-positive GOCs.

Our results demonstrated the importance of local control of brain metastasis with surgery and radiotherapy. Further systemic treatments should be utilised for systemic control, but their role in local control of brain metastasis has not been proven in clinical trials of GOC patients to date. This is an under investigated group; coordinated clinical trials are required to establish the effectiveness of HER2-directed and new novel therapies in HER2-positive GOCs with brain metastasis.

## Figures and Tables

**Figure 1 cancers-14-05754-f001:**
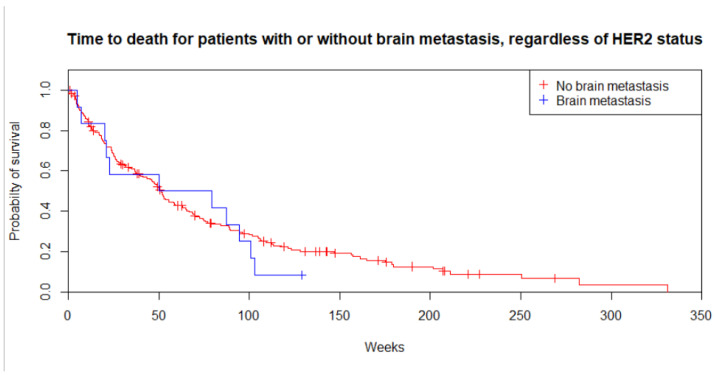
Overall survival for patients with metastatic or recurrent gastric, gastroesophageal junction or oesophageal adenocarcinoma, regardless of HER2 status (*p* = 0.5).

**Figure 2 cancers-14-05754-f002:**
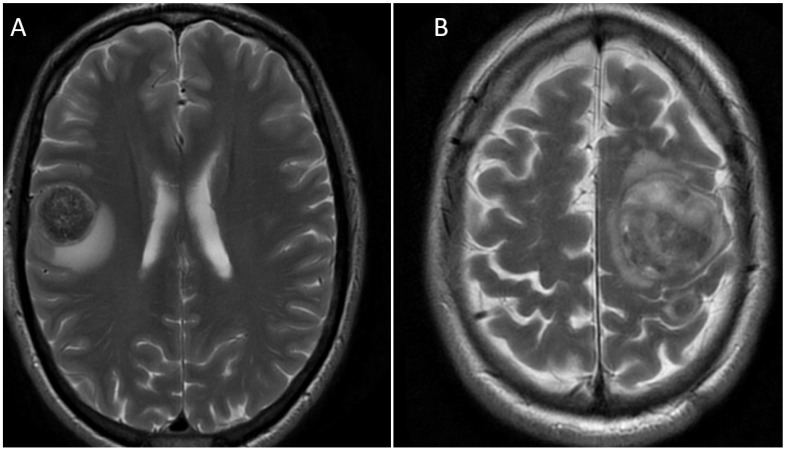
Representative magnetic resonance images (T2 weighted, propeller silent sequencing, and sagittal views of the region of interest) of two patients presenting with metastatic GOCs. (**A**) A 52-year-old male with primary gastroesophageal junction HER2-positive GOC. (**B**) A 50-year-old male with primary gastroesophageal junction HER2-positive GOCs. Arrows denote position of metastases.

**Table 1 cancers-14-05754-t001:** Patient characteristics of metastatic gastroesophageal adenocarcinoma with or without brain metastasis.

Variable	All Patients *n* = 201 (%)	Patients without Brain Metastasis *n* = 189 (%)	Patients with Brain Metastasis *n* = 12 (%)	*p* Value
**Gender** Male Female	162 (80.6) 39 (19.4)	150 (79.4) 39 (20.6)	12 (100) -	0.169
**Age at diagnosis (years)** Mean (SD)	68.9 (12.7)	69.5 (12.4)	58.9 (12.6)	0.005
**Location** Oesophagus Gastro-oesophageal junction Gastric	80 (39.8) 42 (20.9) 79 (39.3)	76 (40.2) 34 (18.0) 79 (41.8)	4 (33.3) 8 (66.7) -	<0.001
**Inclusion criteria** De novo metastatic disease Recurrence post-surgical resection Progression after initially clinically local disease not surgically resected	125 (62.2) 54 (26.9) 22 (10.9)	117 (61.9) 51 (27.0) 21 (11.1)	8 (66.7) 3 (25.0) 1 (8.3)	0.934
**HER2 status** Negative Positive	167 (83.1) 34 (16.9)	162 (85.7) 27 (14.3)	5 (41.7) 7 (58.3)	<0.001

**Table 2 cancers-14-05754-t002:** Clinical presentations and management of patients who developed symptomatic brain metastasis from gastroesophageal primary cancers.

Patient Demographics	Tumour Origin and Initial Staging	Prior Treatments	Presentation of Brain Metastasis	Location of Lesion	Management of Brain Metastasis	Subsequent Therapies	Survival Post-Diagnosis of Brain Metastasis
HER2-positive brain metastasis							
52-year-old, Male	GOJ, local nodal disease, brain metastasis.	Nil, de novo presentation.	Left facial weakness and lower limb sensory changes.	Right parietal lesion.	Stereotactic craniotomy and debulking surgery. Post-operative radiotherapy.	Palliative radiotherapy to GOJ primary. Keynote-811 trial ^a^.	902 days, remains alive at time of data cut off
			Asymptomatic.	Left cerebellar metastasis (6 months post-initial lesion).	Stereotactic craniotomy and debulking surgery. Post-operative radiotherapy.	Continued Keynote-811 trial beyond progression, with addition of Trastuzumab. Phase 1 trial ^b^. Integrate IIB clinical trial ^c^.	516 days. remains alive at time of data cut off
50-year-old, Male	GOJ, local nodal disease, brain metastasis.	Nil, de novo presentation.	Focal seizure, right arm weakness, Expressive dysphasia.	Left frontal, left temporal, right frontal (total 8 lesions).	Excision and biopsy of largest lesion in left frontal. Whole-brain radiotherapy subsequently.	Trastuzumab/Cisplatin/Capecitabine. Palliative radiotherapy to primary. Stereotactic radiotherapy to the left frontal brain lesion on progression. Stereotactic craniotomy and resection of left frontal and parietal lesions. Nil further systemic treatments.	660 days
71-year-old, Male	GOJ, local nodal disease.	Neoadjuvant CROSS. Definitive surgery.	Right sided homonymous hemianopia.	Left occipital lobe mass.	Stereotactic craniotomy and debulking surgery. Post-operative radiotherapy.	Not fit for further treatments.	209 days
56-year-old, Male	GOJ, Liver metastasis.	Palliative radiotherapy to primary tumour. Palliative Trastuzumab/Fluorouracil/Cisplatin.	Persistent nausea and vomiting.	Left parietal and left peduncle.	Stereotactic radiotherapy.	Paclitaxel.	188 days
63-year-old, Male	GOJ, Brain metastasis.	Nil, de novo presentation.	Ataxia, slurred speech.	Innumerable lesions through both hemispheres.	Whole-brain radiotherapy.	Not fit for further treatments.	42 days
75-year-old, Male	Distal oesophageal, liver metastasis.	CAPOX. Palliative radiotherapy to primary tumour. FOLFIRI.	Persistent headache.	Solitary right cerebellar lesion.	Whole-brain radiotherapy.	Not fit for further treatments.	25 days
52-year-old, Male	Distal oesophageal, subcutaneous, intramuscular, and nodal metastasis.	Trastuzumab/Capecitabine/Oxaliplatin. Paclitaxel/Pembrolizumab.	Confusion, disorientation.	Left cerebellar lesion, right parietal, and left thalamus. Extensive leptomeningeal disease.	Palliation.	Not fit for further treatments.	6 days
HER2-negative brain metastasis							
75-year-old, Male	GOJ, local nodal disease.	Neoadjuvant CROSS and total gastrectomy.	Dysphasia, right sided pronator drift.	Right occipital lesion, haemorrhagic.	Palliation.	Not fit for further treatments.	229 days
40-year-old, Male	GOJ, local nodal disease.	Neoadjuvant CROSS and Ivor Lewis oesophagectomy. Bone, liver, and lung recurrence. Chemotherapy ECX, radiotherapy to bone lesion, palliative Paclitaxel.	Partial motor seizure, expressive dysphasia.	Left frontal lesion.	Stereotactic craniotomy and debulking surgery. Post-operative radiotherapy.	Unknown as moved out of area.	223 days
47-year-old, Male	Distal oesophageal, brain metastasis.	Nil, de novo presentation.	Confusion, right upper limb weakness.	Left frontal, smaller lesions left parietal, right post-central gyrus, right frontal.	Stereotactic craniotomy and debulking surgery left frontal lesion. Post-operative stereotactic radiotherapy to surgical bed and other lesions.	Palliative radiotherapy to primary. FLOT chemotherapy.	160 days
75-year-old, Male	Distal oesophageal, lung, liver, cutaneous and intracranial metastasis.	Nil, de novo presentation.	Asymptomatic, identified as part of GP initiated investigation of metastatic disease.	Left parietal, left frontal (small).	Nil.	Capecitabine.	33 days
46-year-old, Male	GOJ, local nodes.	Neoadjuvant FLOT. Radiological progression pre-surgery.	Nausea, dizziness, generalised pain.	Leptomeningeal.	Whole-brain radiotherapy.	Not fit for further treatments.	18 days

^a^ Keynote-811 trial—phase III trial investigating FOLFOX + Nivolumab/placebo. Unblinded to placebo arm. ^b^ GQCT001 Genequantum trial—phase I trial investigating a TDM1-like compound. ^c^ Integrate IIb trial—phase III open-label trial investigating Regorafenib and Nivolumab vs. standard-of-care chemotherapy. Regorafenib and Nivolumab arm. GOJ—gastroesophageal junction; CROSS—chemoradiotherapy for oesophageal cancer followed by surgery study (chemotherapy Carboplatin/Paclitaxel); CAPOX—Capecitabine/Oxaliplatin; FOLFIRI—Fluorouracil/Leucovorin/Irinotecan; ECX—Epirubicin/Cisplatin/Capecitabine; FLOT—Fluorouracil/Leucovorin/Oxaliplatin/Docetaxel.

## Data Availability

Data are available at request through the corresponding author.

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
