# Peer review of "HER2-Positive Gastroesophageal Cancers Are Associated with a Higher Risk of Brain Metastasis"

_cancers, 2022, doi:10.3390/cancers14235754_

Round 1

Reviewer 1 Report

Comments

The study “HER-2 positive gastroesophageal cancers are associated with a 2 higher risk of brain metastasis” is quite interesting. However, the study has many pitfall:

The presentation and language is poor and not reader friendly. The English language need extensive revision, large and complex sentenced to be shortened and simplified, and lucidity in the presentation needs to be extensively improved.

Examples

The identification of metastasis is a devastating event, rendering the disease process 59

 as incurable, and a switch to palliative intent treatment. Improve presentation. Remove as before incurable.

The additional of trastuzumab to a cytotoxic backbone 78

For analysis, we have only included patients where a definitive HER2 results was 130

available.

These were assessed by the Kaplan Meier 140

method. Cite this method in text and provide reference in reference section

There was 167 (83.1%) HER2 157

negative tumours, confirmed by ISH in 47 cases, and by IHC in 120 cases.

Eight (66.7%) of the patients with symptomatic brain metastases presented with 180

metastatic disease at diagnosis, 3 had recurrence following previously resection of the 181

primary disease and the final patient had progression of local disease treated with a non- 182

 surgical approach.

Although small numbers, we observed no statistical difference in survival following 187

 the diagnosis of symptomatic brain metastasis in HER2 positive or negative disease 188

 (median 29.9 weeks vs 22.9 weeks; p=0.6). 189

Make it fluent. What do you mean by small numbers? Is it sample size?

Interestingly both patient’s first site disease progression was subsequent intra-cranial 199

 disease, suggesting a differential effect of the systemic treatments employed.

The devastating effect of brain metastases on functional status is demonstrated with 201

only 5 (41.7%) patients suitable for systemic treatment post diagnosis.

The sentence seems complex. Improve presentation.

The 212 consequence of developing brain metastasis goes beyond patient survival, with significant 213 morbidity associated with their formation and treatment. What the authors mean by "The consequence of developing brain metastasis goes beyond patient survival, with significant morbidity associated with their formation and treatment". It needs further explanation and clarity.

Simplify the sentence. To the best of our 217

 knowledge, this is the first time the significant increased likelihood of developing brain 218

metastasis in HER2 positive GOCs has been demonstrated for patients with 219

adenocarcinomas arising from the oesophagus, gastroesophageal junction, and stomach.

Brain metastasis identified at an earlier stage have improved treatment options 227

and less morbidity associated with their disease. Grammatic check and improve presentation.

This is slightly higher rate than Cavanna et al. [28] which showed a rate of 2.33% 230

 in gastric cancer population, similar to the 6.3% reported by Hejleh et al. in all histology 231 oesophageal cancer [27]. Our cohort of patients,

It should be in our cohort of patients,

prevalence just lower than Hejleh et al. As primary adenocarcinomas from these sites are 234.  it should be as not As.

While 235

previous studies have shown brain metastasis associated with HER2 positivity in GOCs 236

[27, 28], we have demonstrated the increased risk of the whole population of recurrent 237

and metastatic GOCs. Improve presentation.

All bar one 242

patient who developed brain metastasis was male in our study; our frequency of males in 243

 this cohort and developing brains metastasis is similar to other brain metastasis from GOC 244

cohorts [30].

our frequency of males in 243

 this cohort and developing brains metastasis is similar to other brain metastasis from GOC 244

cohorts [30]. What do you mean by our frequency? What authors mean by developing brains?

Although small 247

numbers, our results show the importance of including adequate local treatments in the 248

What authors mean by small numbers. Improve the presentation.

Novel biomarker targeted therapies have been 249

demonstrated to improve cancer outcomes, and in melanoma has potentially removed the 250 necessity of surgery in the management of brain metastasis [31]. Due to the low numbers 251

of brain metastasis in GOCs, large trials have proven difficult to develop to clarify the 252

optimum treatment paradigm for these patients

Surprisingly, lapatinib a small molecule tyrosine kinase 276

inhibitor (TKI) against epidermal growth factor receptor and HER2, only showed benefit 277

in treatment naïve patients in combination with capecitabine, and did not demonstrate 278

radiological or survival benefits otherwise, in spite of its increased ability to cross the 279

blood brain barrier [34].

Authors have mentioned that Fluoropyrimadines, platinums, taxanes and topoisomerase I 69

inhibitors have all been shown to improve overall survival in metastatic GOC [16]. 70 Furthermore, recent phase III data has demonstrated the additional benefits of 71 immunotherapy with checkpoint inhibitors such as Nivolumab and Pembrolizumab, both 72 in combination with chemotherapy and as monotherapy [17-20]. Clinical biomarkers in 73 routine practice are the human epidermal growth factor receptor 2 (HER2 or ERBB-2 74 receptor), program death- ligand 1 (PD-L1) and DNA mismatch repair genes [18, 21, 22]. 75 Trastuzumab is a monoclonal antibody targeting HER2 preventing signalling mediated 76 by the HER2 through the phosphatidylinositol 3-kinase (PI3K) and mitogen-activated 77 protein kinase (MAPK) cascades. The authors are encouraged to discuss about the properties of these compounds-Fluoropyrimadines, platinums, taxanes and topoisomerase I 69

inhibitors, Nivolumab and Pembrolizumab and  Trastuzumab that render them useful therapeutic agents.

We performed a retrospective review of patients treated for gastric, gastroesophageal 88 junction or oesophageal (GOC) adenocarcinomas within the Illawarra Shoalhaven Local 89 Health District (ISLHD), New South Wales, Australia. Why only previously treated persons and why not newly diagnosed persons.

Why the study was single centred?

Provide the flow chart showing how you recruited patients.How many agreed to participate and how many were reluctant.?. Have you obtained informed consent from all participants. The information is missing.

What are early symptoms at which the patients should be diagnosed for GOC-induced brain metastasis so as to prevent its progression?

What in view of the authors is the most effective therapies for GOC-induced brain metastasis ?

HER2 immunohistochemistry (IHC) and In Situ 113 Hybridization (ISH) was performed on patients. Fluorescent microscopic visuals of tissues studied with immunohistochemistry are lacking and need to be provided. As well as the microscopic visuals of tissues must be provided for In Situ Hybridization (ISH).

The computer tomography (contrast) and magnetic resonance images are not provided. The images must be provided to support your observation.

Authors have not mentioned whether HER2 positive breast cancer patients have been excluded from the study.

CSF via surgically placed reservoirs or intrathecal 259. Provide fullform of CSF

Check overall survival. Authors must use fullform along with short form in parenthesis in first instance and then only short form all through the manuscript.

These were assessed by the Kaplan Meier 140

method. Cite this method in text and provide reference in reference section

Statistical analysis was performed using R version 4.0.2. Descriptive summary 146

Put the version in parentheses

 Overall the manuscript needs substantial revision

Author Response

Please see attached word document

Reviewer 2 Report

This manuscript has been described Her-2 positive gastroesophageal cancers are associated with a higher risk of brain metastasis comparing HER-2 negative ones. Unfortunately, this manuscript could not contribute decision of treatment strategies for readers. 

Almost patients with brain metastasis had multiple metastases in other organs. This means brain metastasis is shown in far advanced phase, not brain specific. HER-2 positive cancers have higher progression potential than HER-2 negative ones. However, Trastuzumab treatment can improve their poor prognosis. In this series, 10 of 12 patients with brain metastasis did not receive trastuzumab treatment prior to definition of brain metastasis. As author mentioned, trastuzumab fails to cross the blood brain barrier, trastuzumab should be used in early treatment phase. Although the incidence of brain metastasis is statistically significant higher in GOCs patients with HER-2 positive than those without HER-2, this cohort is very small and single institutional assessment.

Author Response

Please see attached word document. 

Reviewer 3 Report

In this work, authors reported a retrospective study on patients with gastroesophageal cancers, especially assessing the risk of HER2 expression for developing brain metastasis. This works is potentially interesting, providing some clinical information in the field. However, there were several issues needed to be addressed.

1.    The second paragraph about the identification, detection of metastasis in the introduction section is not directly relevant to this work. Authors can discuss metastatic patterns in GOCs, potential prognostic factors for metastases, and clinical outcomes.

2.    The third paragraph about the chemotherapy is not relevant to this work, authors can discuss about the HER2 biology in GOCs, and current studies about HER2 in cancer metastasis.

3.    For the GOCs spelling, author should keep it same across the manuscript, and GOC at lines 66, 69, 70, 106 etc.. need to be corrected.

4.    For HER2 detection, as a scientific work, authors need to provide HER2 IHC or FISH imaging results at least in the supplementary.

5.    In table 1, author only provided percentage values, a bar plot with a test of statistical significance is need to support author’s hypothesis that the incidence of brain metastasis was significantly higher in patient with HER2 positive tumors.

Author Response

Please see attached word document. 

Round 2

Reviewer 1 Report

The authors have improved the manuscript. However, still we have the following concerns:

Check the grammar of this sentence. “For analysis, we have only included patients where a definitive HER2 results was 152

available”.

The 242 consequence of developing brain metastasis goes beyond patient survival, with significant 243 neurological complications associated with brain metastasis formation and treatment. 244.

What are the consequences of brain metastasis goes beyond patient survival. It seems a confusing. It needs further explanation and clarity.

Check the grammar of this sentences “The issue with these studies are they did 252

 not assess all patients eligible for HER2 directed therapies and were unable to advise on 253

 the increased risk of developing brain metastasis in HER2 positive GOCs.”

It is all right that the authors have included the details where the testing was carried out but authors are again encouraged to include Fluorescent microscopic visuals of tissues studied with immunohistochemistry as well as the microscopic visuals of tissues for In Situ Hybridization (ISH), since there are no ethical issues associated if included.

The MRIs seem to be of different views. Can you mention what each figure is in which view- axial, sagittal or coronal. “Figure 2. Representative magnetic resonance images (T2 weighted) of two patients presenting with 236 metastatic GOCs. A) 52-year-old male with primary gastroesophageal junction HER2 positive GOC 237 B) 50-year-old male with primary gastroesophageal junction HER2 positive GOCs. Arrows denote 238 position of metastases”.

Author Response

Please see the attached word document. 

Reviewer 3 Report

Authors had addressed all comments.

Author Response

Thank you for your review.